# Avapritinib-based SAR studies unveil a binding pocket in KIT and PDGFRA

A. Teuber [1,5], T. Schulz [1,5], B. S. Fletcher[2], R. Gontla [1], T. Mühlenberg [2], M.-L. Zischinsky[3], J. Niggenaber [1], J. Weisner [1], S. B. Kleinbölting[1], J. Lategahn [1], S. Sievers [4], M. P. Müller [1], S. Bauer [2] & D. Rauh [1] ✉

Avapritinib is the only potent and selective inhibitor approved for the treatment of D842V-mutant gastrointestinal stromal tumors (GIST), the most common primary mutation of the platelet-derived growth factor receptor α (PDGFRA). The approval was based on the NAVIGATOR trial, which revealed overall response rates of more than 90%. Despite this transformational activity, patients eventually progress, mostly due to acquired resistance mutations or following discontinuation due to neuro-cognitive side effects. These patients have no therapeutic alternative and face a dismal prognosis. Notable, little is known about this drug's binding mode and its medicinal chemistry development, which is instrumental for the development of the next generation of drugs. Against this background, we solve the crystal structures of avapritinib in complex with wild-type and mutant PDGFRA and stem cell factor receptor (KIT), which provide evidence and understanding of inhibitor binding and lead to the identification of a sub-pocket (Gα-pocket). We utilize this information to design, synthesize and characterize avapritinib derivatives for the determination of key pharmacophoric features to overcome drug resistance and limit potential blood-brain barrier penetration.

Gastrointestinal stromal tumors are the most common sarcomas and most common mesenchymal tumors of the gastrointestinal tract. The identification of KIT and PDGFRA activating mutations[1,2] as the central oncogenic drivers in GIST has transformed GIST from an untreatable, highly fatal disease to a paradigmatic disease for the impact of targeted treatments, such as imatinib[3–5]. Therefore, GIST is a beacon of precision oncology, demonstrating the transformative impact of personalized treatments for unique genetic abnormalities in cancer therapy. While imatinib is highly potent against KIT exon 11 mutant GIST and also shows activity against some PDGFRA activation loop (AL) mutations, the PDGFRA-D842V mutation has for more than a decade been notoriously unresponsive to any known kinase inhibitor[3,4,6–8]. By favoring the active kinase conformation, the D842V mutation within

the AL leaves type II inhibitors such as imatinib, which bind to inactive kinase conformations, ineffective (Fig. 1)[9,10]. The treatment options for these patients have only very recently dramatically changed following the approval of avapritinib[8]. Avapritinib (**1**), an ATP competitive inhibitor, binds to the active kinase conformation and was designed to inhibit PDGFRA-D842V and iso-structurally mutated KIT-D816V/H or similar mutants of the AL[11,12]. Due to its high potency toward AL mutant KIT, avapritinib also has great clinical relevance in advanced systemic mastocytosis[13]. However, while the overall toxicity during avapritinib treatment is manageable, cognitive side effects are a concern in a subset of patients, particularly when risk-mitigation strategies, such as early treatment interruptions, are not followed. These effects most likely result from avapritinib's ability to penetrate the blood-brain

[1]Department of Chemistry and Chemical Biology, TU Dortmund University and Drug Discovery Hub Dortmund (DDHD), Zentrum für Integrierte Wirkstoffforschung (ZIW), Otto-Hahn-Strasse 4a, 44227 Dortmund, Germany. [2]Department of Medical Oncology and Sarcoma Center and West German Cancer Center, DKTK partner site Essen, German Cancer Consortium (DKTK), University Duisburg-Essen, Medical School, Essen, Germany. [3]Lead Discovery Center GmbH, Department for in vitro ADME and PK, Otto-Hahn-Strasse 15, 44227 Dortmund, Germany. [4]Compound Management and Screening Center, Max Planck Institute of Molecular Physiology, Dortmund, Germany. [5]These authors contributed equally: A. Teuber, T. Schulz. ✉e-mail: daniel.rauh@tu-dortmund.de

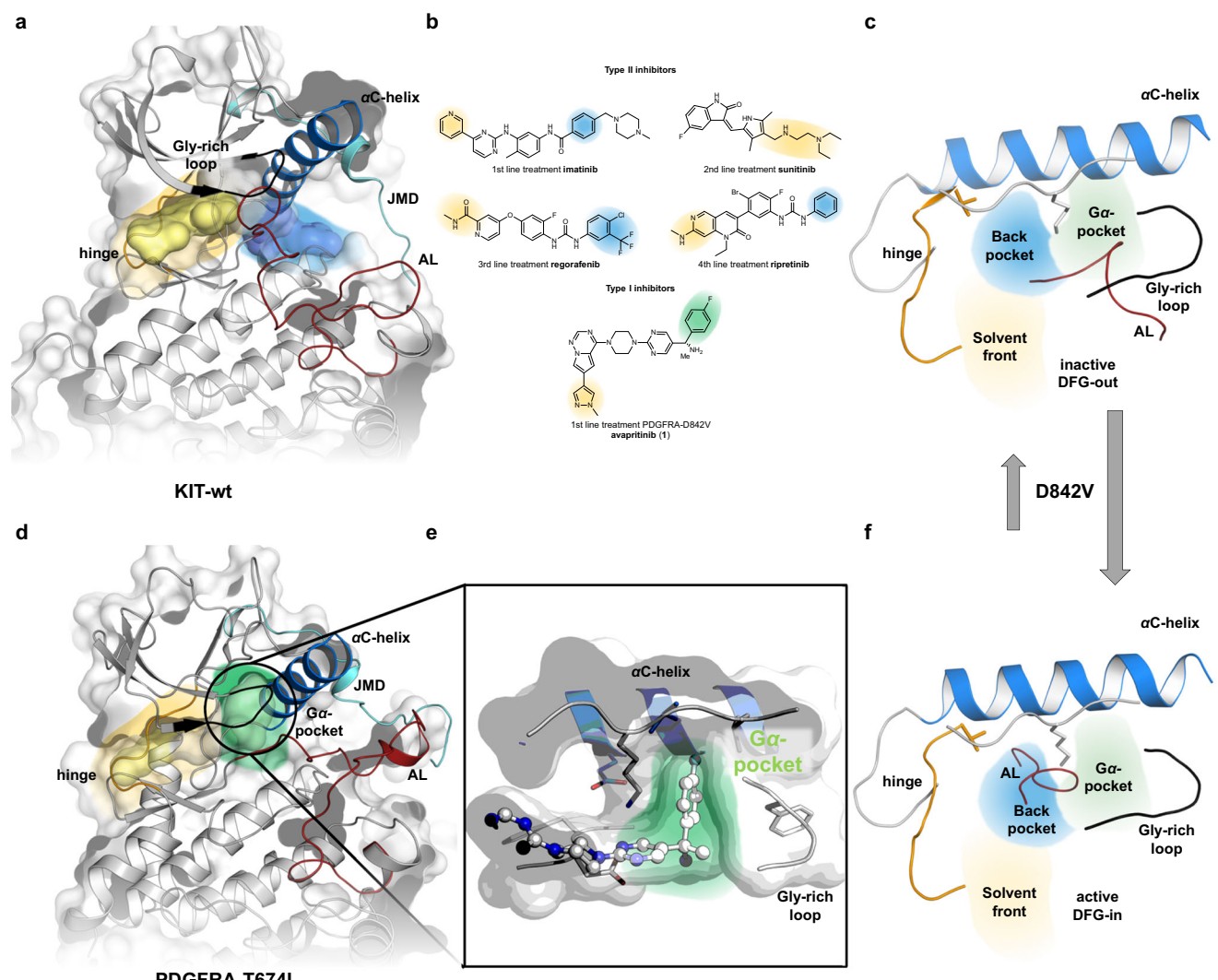

**Fig. 1 | Crystal structures of type I and type II inhibitors used in GIST treatment bound to target proteins KIT and PDGFRA. a** Overview of the DFG-out conformation of the kinase domain bound to type II inhibitors. Inhibitors imatinib (PDB-ID: 1T46), sunitinib (PDB-ID: 3G0E) and a ripretinib derivative (PDB-ID: 6MOB) are shown as surfaces, illustrating their binding modes in different pockets of the protein. **b** Two-dimensional representation of the inhibitors approved for GIST treatment. **c, f** Schematic representation of the inactive DFG-out (**c**) and the active DFG-in (**f**) kinase conformations, reveiling that a D842V-mutation in PDGFRA would shift the equilibrium toward the active DFG-in conformation. **d** Overview of the obtained co-crystal structure of PDGFRA-T674I bound to avapritinib (**1**) in the DFG-in conformation (PDB-ID: 8PQH). **e** Highlighting the Gα-pocket addressed by avapritinib.

barrier (BBB) and interact with unidentified targets in the brain[8,12]. Patients who progress on avapritinib and then discontinue treatment face a highly dismal outcome with a median overall survival of only several weeks[9]. No approved drug currently shows activity against D842V primary mutants or against clones that have double-mutant compound mutations that emerge upon avapritinib treatment.

Unfortunately, the data on the medicinal-chemical development of avapritinib available in the public domain is rather limited. Particularly lacking is structural information that could help to better understand the sensitivity of avapritinib to resistance mutations or the structural features that could be modified to block BBB penetration. In the absence of experimental structural data, various chemoinformatics approaches have been employed to evaluate the binding mechanism of avapritinib to its target kinase. However, these methods have produced disparate conclusions[11,14]. Gardino et al. provided brief insights into the development of avapritinib and presented a docking complex by using a hybrid structure based on a crystal structure of imatinib bound to KIT in an inactive conformation (PDB-ID: 1T46), replacing the AL with that of a KIT crystal structure in an

active conformation (PDB-ID: 1PKG)[15]. Gilreath et al. speculated on the exact binding mode based on those of type I inhibitors with related kinases and highlighted the lack of published data on how avapritinib binds to its kinase targets[14]. Winger et al. used molecular docking with an active conformation of KIT-D816V to investigate the resistance profile of avapritinib in mutant KIT, particularly the gatekeeper mutation T670I[16]. Their proposed binding mode includes the pyrrolo[2,1-f][1,2,4]triazine scaffold nitrogen bound to the hinge region's Cys673 carbonyl. Grunewald et al. identified a diverse set of resistance mutations emerging in PDGFRA during avapritinib treatment, including the solvent-front mutation G680R as well as the gatekeeper mutations T674I/R[9,17]. Our group contributed computational analyses to this study, including docking avapritinib into a homology model exhibiting the active conformation of PDGFRA-D842V, which was validated by molecular dynamics (MD) studies demonstrating the binding mode and suggesting a molecular rationale for drug-resistant mutations[9]. However, the abovementioned studies were partly contradictory. An experimental structural model is urgently needed to unequivocally resolve the binding pose of avapritinib to

KIT/PDGFRA and to understand and characterize the effects of resistance mutations.

Therefore, we set out to address this question, and we successfully crystallized KIT and PDGFRA variants in complex with avapritinib to gain deeper insights into ligand binding. With these insights into the exact mode of binding of avapritinib, we subsequently characterized the contributions of different parts of the molecule to the potency of the drug. We also identified potential avenues for further improvement, including means to obtain a better profile in terms of resistance mutations and possible ways to avoid adverse effects due to penetration of the blood-brain barrier.

## Results

### X-ray crystallography reveals specific interactions

We aimed to complement and validate the in silico analyses[9] with experimental data. We set out to reveal the binding mode of avapritinib and gain a better understanding of the resistance mechanisms by means of protein X-ray crystallography. Against this background, attempts have been made to establish expression and crystallization conditions for several clinically relevant mutants of KIT and PDGFRA. However, to date, these attempts only resulted in the successful expression of stable and active wild-type (wt) KIT and PDGFRA proteins as well as gatekeeper mutants. The currently available proteins have been selected as model systems and subjected to crystallization studies. The structures obtained provide valuable insights and also allow us to deduce the effects conferred by other resistance mutations. In total, we obtained protein crystals suitable for structure determination which resulted in a total of 12 crystal structures: two apo-crystal structures and ten complex structures of KIT-wt, -T670I, and PDGFRA-T674I bound to avapritinib and to different ligands synthesized in our laboratory.

The complex crystal structure of avapritinib bound to PDGFRA-T674I (Fig. 1d) demonstrated, as anticipated by Grunewald et al., that avapritinib binds an active-like kinase conformation, which is characterized by an extended AL and a DFG-in state as was evident from Asp836 (or Asp810 in KIT) of the Asp-Phe-Gly (DFG) motif pointing into the ATP-binding site (Fig. 1d, f). Avapritinib interacts with the hinge region via H-bonding with the backbone amine of Cys677 via the pyrrolotriazine scaffold N2 nitrogen. The scaffold further interacts with Leu599, Leu825, Ala625, and Tyr676. The solvent-exposed methyl pyrazole undergoes a cation-π-interaction with Arg597. The linking piperazine forms contacts with Val607 and Cys835, while the pyrimidine interacts with Val607 and forms an additional hydrogen bond with the catalytic Lys627. For the primary amine located at the stereocenter, we observed an ionic interaction between the primary amine and Asp836 from the DFG motif (Supplementary Figs. 1a, b, and 2a).

The binding of avapritinib stabilizes the DFG-in conformation of the kinase domain in which the amino acids Gly605 and Val607 of the Gly-rich loop, Leu641 of the αC-helix, the catalytic Lys627 and Leu629 form a hydrophobic pocket that can perfectly accommodate the shape of the fluorobenzene moiety of the inhibitor. Additional cation-π interactions of the fluorobenzene were observed with the catalytic Lys627. This pocket extends beyond the phosphate-binding region of the ATP-binding site and is distinct from the backpocket (often referred to as the switch or deep pocket in some publications[18–20]) and has not been targeted or studied previously in the field of KIT and PDGFRA or related kinases; therefore, we refer to this structural feature as the Gα-pocket (Fig. 1e), to account for its location between the αC-helix and Gly-rich loop.

Next, we compared the obtained crystal structure with publicly available structures of KIT and PDGRA in inactive (PDB-IDs: 1T46, 4U0I, 6MOB) and active (PDB-IDs: 1PKG, 7KHK, 7KHJ) conformations. Interestingly, we observed that the Gα-pocket addressed by avapritinib appears to manifest only upon ligand binding. Both in the active state and several inactive conformations, this pocket does not seem to be accessible. On the one hand, in the active state, the Gly-rich loop and the αC-helix are rotated inward into the ATP-binding site so that the Gα-pocket does not appear present. On the other hand, in the inactive state, the pocket is occupied by the AL or, in particular, Leu839 of the DFGLARDI motif. Avapritinib mimics this key feature, thereby displacing the AL and forcing an αC-helix-out yet DFG-in conformation (Supplementary Fig. 3). To our knowledge, the Gα-pocket has not been targeted by other reported inhibitors in the field of GIST and has only been observed upon the binding of avapritinib and corresponding derivatives. It is yet to be determined whether an induced fit or conformational selection mechanism leads to the formation of the Gα-pocket. However, we assume that this pharmacophore significantly contributes to the activity and selectivity of avapritinib. Thus, our detailed analysis showed that the binding of avapritinib to a DFG-in, but αC-helix-out conformation characterizes it as a type 1.5 inhibitor rather than a canonical type 1 inhibitor[21]. Following these findings, we analyzed different states of KIT and PDGFRA, including ligand-bound and apo-crystal structures, with a particular focus on analysis of the assembly of the R-spine (Supplementary Fig. 4). In the inactive DFG-out conformations (apo, PDB ID: 8PQJ, type II inhibitor bound, PDB ID: 1T46), an absence of R-spine formation was observed, as expected (Supplementary Fig. 4d, e). Conversely, in the activated ATP-bound state (PDB-ID: 1PKG), R-spine assembly was observed (Supplementary Fig. 4c). Interestingly, the co-crystal structures of avapritinib revealed that the R-spine is also able to assemble (Supplementary Fig. 4a, b). These conformational findings, which only became evident with the analysis of the obtained crystal structures, will be of great importance for the design and development of next-generation inhibitors to overcome acquired drug resistance and it highlights the importance of high-resolution complex crystal structures in drug discovery.

### Structural insights into resistance mechanisms

Previously, we found that mutation of the gatekeeper PDGFRA-Thr674 to a non-polar isoleucine (T674I) or a sterically more demanding arginine (T674R) induces resistance to avapritinib[9]. In order to clarify the impact of these mutations, we set out to solve a pair of complex crystal structures. Intense screening and optimization resulted in two complex structures with avapritinib in wt and gatekeeper-mutant KIT-T670I (Fig. 2a, b). To the best of our knowledge, this is the first report of a structure of T670I mutated KIT. As anticipated, a water-mediated interaction of Thr670 of KIT-wt and the N4 of the pyrrolotriazine scaffold is clearly visible in this structure (Fig. 2a). In contrast, in the T670I-mutated kinase, this position is occupied by the isoleucine side chain, and no water-mediated contact can be observed (Fig. 2b, Supplementary Fig. 2d, e). Similar results were also obtained for apo-PDGFRA-wt and -T674I, showing that the water molecule is located near the gatekeeper in the case of the PDGFRA-wt structure but not in the PDGFRA-T674I structure (Fig. 2c, d). The water molecule of the wild-type structure at this position is also conserved in several other published X-ray structures of KIT and PDGFRA (PDB-IDs: 1T45, 6GQK, 6GQL, 6GQM). In contrast to the published working hypothesis[16], the water molecule thus forms a polar contact with Thr670 instead of a water network involving the catalytic lysine and the gatekeeper mutation, which results in a loss of an important interaction. Additionally, the introduced hydrophobic isoleucine is close to the polar scaffold, presumably contributing to a reduced binding affinity. Based on their MD simulation, Winger et al. suggested that the T670I mutation confers resistance via distant conformational changes within the Gly-rich loop, which we cannot confirm based on our complex crystal structures. However, we acknowledge that an additional effect, such as increased ATP affinity, cannot be excluded[16,22].

With respect to the second described resistance mutation (T674R), loss of activity is likely caused by steric repulsion with the sterically demanding arginine residue analogous to many described

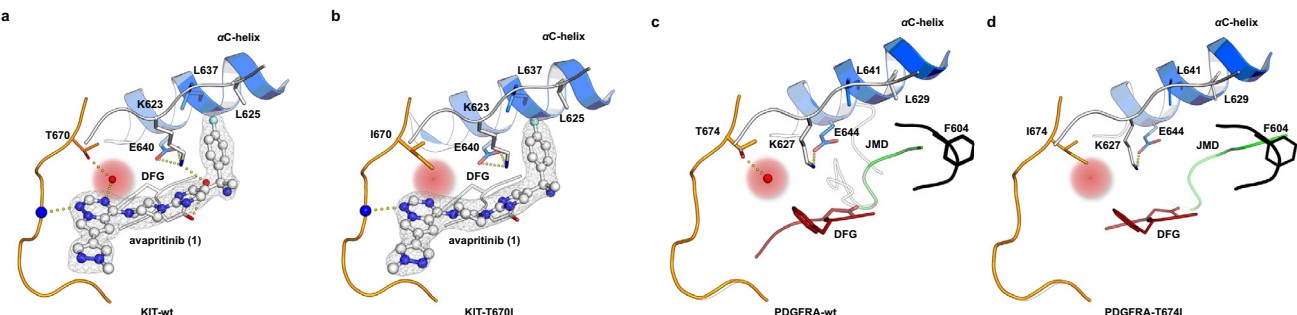

**Fig. 2 | Structural analysis of the gatekeeper mutation occurring during avapritinib treatment. a** Complex crystal structure of **1** bound to KIT-wt (PDB-ID: 8PQ9). Herein a water-mediated interaction between the N4 of the pyrrolotriazine scaffold of **1** and Thr670 is clearly visible in this structure. **1** is shown with its |2F$_o$-Fc|· electron density map at an r.m.s.d. = 1.0. **b** Complex crystal structure of **1** bound to KIT-T670I (PDB-ID: 8PQG). It is seen that the isoleucine of Ile670 replaces a water molecule observed in the wt structures, leading to drug resistance. **1** is shown with its |2F$_o$-Fc|· electron density map at an r.m.s.d. = 1.0. **c** Apo-crystal structure of

PDGFRA-wt showing an auto-inhibited DFG-out kinase conformation with the JMD (green) protruding into the backpocket of the protein. In addition, the structure reveals that a water molecule is bound to the gatekeeper amino acid Thr674 (PDB-ID: 8PQJ). **d** Apo-crystal structure of PDGFRA-T674I showing an auto-inhibited DFG-out kinase conformation with the JMD (green) protruding into the backpocket of the protein. Furthermore, the interaction between a water molecule and the gatekeeper amino acid Ile674 cannot be observed and the water molecule is displaced (PDB-ID: 8PQK).

gatekeeper mutations. The obtained structures support the resistance mechanism of the prominent G680R mutant already postulated by Grunewald et al. Although there is no direct interaction between avapritinib and Gly680, the structures are in proximity by about 4 Å, suggesting the mutation to arginine introduces a steric clash with the sterically more demanding guanidinium group (Supplementary Fig. 5). In the cases of the N659K and V658A (equivalent to V654A in KIT) mutations, which are located near the αC-helix, these mutations may lead to a loss of stabilizing effects between important regulatory elements of the kinase domain, favoring a conformation less suitable for efficient ligand binding (Supplementary Fig. 5). Accordingly, it can be concluded that all described mutations confer resistance either via local effects near the hinge region (e.g., V658A, T674I, G680R) or via global effects that shift the equilibrium toward the active conformation (e.g., AL, etc.).

Finally, comparing the binding modes of avapritinib in PDGFRA and KIT, it is striking that the interaction of the catalytic lysine (Lys627 in PDGFRA, Lys623 in KIT) with the pyrimidine was not observed in KIT (Supplementary Figs. 1d and 2a–c). Additional water-mediated interactions were identified in KIT linking the ligand's primary amine and Lys623 and Asp810 (Supplementary Figs. 1e and 2b, c). Apart from this, the mode of binding is very similar, and almost all interactions described for PDGFRA are also evident in KIT. Therefore, it may be expected that resistance mechanisms initially found in either KIT or PDGFRA can also occur in the other kinase, respectively.

## Structure-guided design to explore important pharmacophoric features

To gain an even deeper understanding of how to overcome resistance mutations, we have chemically modified key elements of the core structure of avapritinib to study their influence on biological activity (Fig. 4a). First, since the solved crystal structures of avapritinib in KIT-wt, - T670I and PDGFRA-T674I highlighted the importance of the Gα-pocket, we chose a deconstruction approach to elucidate further the importance of the fluorobenzene moiety addressing this sub-pocket (Fig. 4a, Supplementary Fig. 6a). Second, instead of the piperazine, heterocyclic spiro-based linkers were investigated. This was done to test different linker lengths connecting the hinge binding element to the Gα pocket moiety, as well as to explore their chemical nature as complex, highly saturated structures with a significant three-dimensional character. (Fig. 4a, Supplementary Fig. 6a). The crystal structures obtained during this work revealed two suitable solvent-exposed exit vectors that would qualify for the modification of

avapritinib to potentially block BBB penetration without negatively affecting potency: (1) the solvent-exposed methyl-pyrazole and (2) the primary amine located at the stereocenter. In this study, we focused on the derivatization of the amine to investigate its interaction with the DFG motif and to modify physicochemical properties such as the molecular weight (MW) or measure for hydrophilicity (logP). For the deconstruction, nucleophilic aromatic substitutions were conducted based on the conversion of the commercially available 4-chloro-6-(1-methyl-1H-pyrazol-4-yl)pyrrolo[2,1-f][1,2,4]triazine **2** with either 1-Boc-piperazine, followed by acid-catalyzed Boc-deprotection, or a nucleophilic substitution with 2-(piperazin-1-yl)pyrimidine yielding compounds **3** and **4**, respectively (Supplementary Fig. 6a). For the synthesis of the spiro-containing compounds, a four-step synthesis route was developed and established (Supplementary Fig. 6a). Starting with a nucleophilic substitution of the triazine scaffold to incorporate various Boc-protected heterocyclic spiro moieties, followed by the carbamate cleavage, nucleophilic substitution with 5-bromo-2-chloropyrimidine, and finally Suzuki coupling with 2-benzyl-4,4,5,5-tetramethyl-1,3,2-dioxaborolane, we obtained **5-7**. To prepare a focused set of amine derivatives, we applied two different alkylation procedures and three amidation late-stage functionalizations on avapritinib, resulting in compounds **8-13** (Supplementary Fig. 6b). The successfully synthesized avapritinib-based inhibitors as well as pyrrolo-triazine **2** were biochemically and cellularly evaluated against mutant forms of KIT and PDGFRA (Fig. 3, Table 1, Supplementary Table 1). The biochemical evaluations were performed using an activity-based homogenous time-resolved fluorescence (HTRF) assay of PDGFRA-D842V and KIT-D816H to determine initial structure-activity-relationships (SAR). Additional profiling procedures were conducted on patient-derived or CRISPR/Cas9-generated isogenic GIST cell line models for PDGFRA (T1-a-D842V, T1-a-G680R, and T1-a-T674I/R) and KIT (GIST-T1, T1-D816E, T1-T670I) as well as GIST-48B (Fig. 3, Table 1, Supplementary Tables 1 and 12). The latter is neither KIT nor PDGFRA dependent, thus serving as a control for off-target toxicity (Table 1, Supplementary Table 12). To further analyze off-target toxicity, breast cancer cell lines ZR-75-1, MDA-MB-175VII, not driven by KIT and PDGFRA, and kinase-independent leiomyosarcoma cell line SK-LMS-1 were also investigated (Supplementary Tables 2 and 12).

## Deconstructing avapritinib highlights the importance of the Gα-pocket

The hinge binding element **2** itself had a rather high IC$_{50}$ (>7000 nM for PDGFRA-D842V), and the potency was not significantly improved

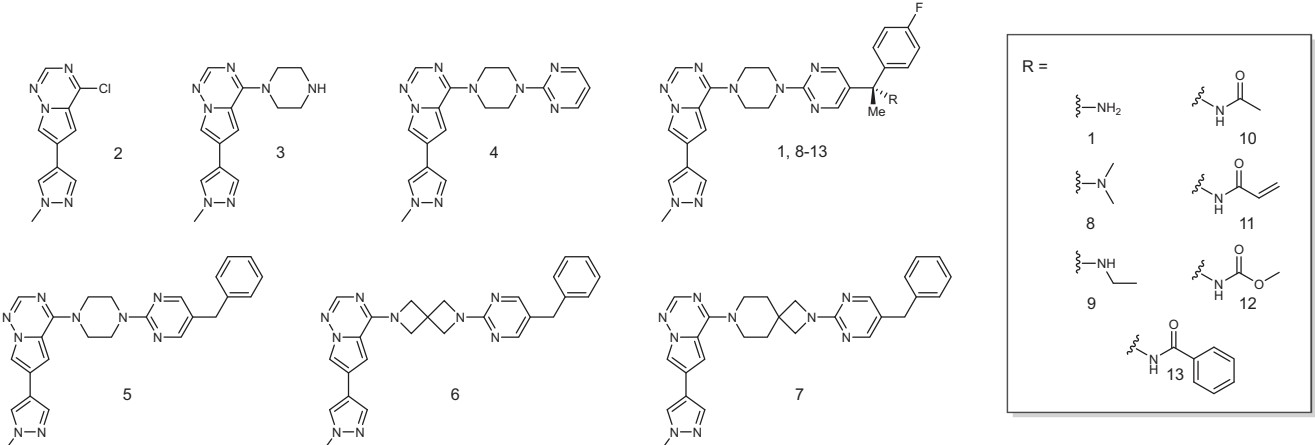

**Fig. 3 | Overview of avapritinib (1) and avapritinib-based inhibitors 2-13 for biological evaluation.** Ligands 1-13 have been characterized both biochemically and cellularly in selected kinases and cell systems in order to evaluate their potencies.

**Table 1 | IC$_{50}$ and GR$_{50}$ determinations on different PDGFRA- and KIT-mutants and GIST cell lines of ligands 1-13**

| Cpd | HTRF IC$_{50}$ [nM] | | KIT CTG GR$_{50}$ [nM] | | PDGFRA CTG GR$_{50}$ [nM] | | | |
| --- | --- | --- | --- | --- | --- | --- | --- | --- |
| | PDGFRA-D842V | KIT-D816H | GIST-T1[a] | T1-D816E | T1-a-D842V | T1-a- G680R | GIST-48B[b] | ratio GIST-48B/ T1-a-D842V |
| ava (1) | <0.1 | 0.5 ± 0.1 | 36 ± 10 | 67 ± 18 | 15 ± 8 | 1563 ± 241 | 1340 ± 173 | 89 |
| 2 | 7324 ± 2078 | 9303 ± 1208 | 6731 ± 2073 | 9481 ± 1038 | 2733 ± 517 | 2000 ± 403 | 7967 ± 2792 | 3 |
| 3 | 5588 ± 2557 | 1541 ± 852 | ≥10,000 | ≥10,000 | ≥10,000 | ≥10,000 | ≥10,000 | 1 |
| 4 | 35 ± 13 | 76 ± 45 | 5271 ± 509 | 8739 ± 1321 | 1887 ± 545 | ≥10,000 | ≥10,000 | 5 |
| 5 | 0.6 ± 0.1 | 1.5 ± 1.0 | 1484 ± 1203 | ≥10,000 | 163 ± 47 | ≥10,000 | ≥10,000 | 61 |
| 6 | 7351 ± 884 | 2158 ± 167 | ≥10,000 | ≥10,000 | ≥10,000 | ≥10,000 | ≥10,000 | 1 |
| 7 | 749 ± 142 | 204 ± 84 | 709 ± 196 | 1962 ± 443 | 8216 ± 1475 | ≥10,000 | ≥10,000 | 1 |
| 8 | 0.4 ± 0.1 | 2.3 ± 1.7 | 254 ± 74 | 515 ± 94 | 58 ± 13 | 2600 ± 384 | 1933 ± 472 | 33 |
| 9 | <0.1 | 0.4 ± 0.2 | 92 ± 16 | 235 ± 51 | 26 ± 5 | 2767 ± 459 | 4033 ± 653 | 155 |
| 10 | <0.1 | 0.5 ± 0.1 | 39 ± 6 | 86 ± 19 | 26 ± 21 | 3311 ± 534 | 4833 ± 2564 | 186 |
| 11 | <0.1 | 0.5 ± 0.2 | 39 ± 7 | 89 ± 23 | 24 ± 6 | 4033 ± 876 | 3467 ± 866 | 144 |
| 12 | 0.2 ± 0.1 | 0.6 ± 0.4 | 60 ± 12 | 131 ± 37 | 19 ± 6 | 3167 ± 58 | 4867 ± 1415 | 256 |
| 13 | 0.2 ± 0.1 | 2.0 ± 0.1 | 287 ± 176 | 637 ± 111 | 51 ± 7 | 5729 ± 435 | ≥10,000 | 196 |

Data presented as mean values ± s.d; $n \geq 3$, where $n$ represents the number of independent experiments.
T1-a cell lines: PDGFRA cell lines based on GIST-T1, generated by CRISPR/Cas9-mediated gene editing. Ava: avapritinib as reference inhibitor.
*n.d.* not determined, *IC50* half-maximal inhibitory concentration, *GR50* half-maximal growth inhibition rate.
[a]GIST-T1 (RRID:CVCL_4976, Val560_Tyr578del).
[b]Control cell line, which is neither KIT nor PDGFRA dependent and serves as an indicator for off-target toxicity.

by the introduction of piperazine (3) (Fig. 3, Table 1). However, further decoration with a pyrimidine (4) resulted in a 160-fold increase in potency with respect to PDGFRA-D842V (Fig. 3, Table 1). When compared to avapritinib, it was clear that the attachment of the fluorobenzene, which binds the Gα-pocket, was 150–350-fold more potent than 4 with respect to KIT-D816H and PDGFRA-D842V, respectively (Fig. 1, Table 1). Hence, as anticipated, the deconstruction series showed a decrease in potency with decreasing complexity, both for PDGFRA and KIT (avapritinib < 4 < 3 ≤ 2). This result is particularly evident in the cellular context, where only 4 and avapritinib retained potency, which is likely due to the omission of essential protein-ligand-interactions in the smaller fragments, particularly those of the stereocenter and the Gα-pocket binding moiety. Overall, the data indicate that binding to the Gα-pocket is an essential structural element for potent ligand binding. To gain structural insight into the formation of the Gα-pocket, the compounds were subjected to our crystallization program, which yielded a high-resolution crystal structure of KIT-wt in complex with 4. The complex structure of 4 bound to KIT-wt revealed that the Gα-pocket is covered by Phe600 of the Gly-rich loop instead of the fluorobenzene (Fig. 4b).

In summary, our biochemical, cellular, and structural data suggest an important contribution of targeting the Gα-pocket in order to obtain potent inhibitors for KIT and PDGFRA. Since currently known resistance mutations are mostly located near or within the hinge region, targeting the remote Gα-pocket presumably provides avenues to overcome the limitations of these known acquired resistance mutations.

## Piperazine as a linking element to lock the compound conformation

Compared to avapritinib, 5 mainly differs in the linking carbon between the pyrimidine of the eastern part and the Gα-pocket binding element. This structural feature allows the investigation of the stereocenter's influence regarding its role in the correct positioning of the eastern part for binding to the protein and the resulting potency. We would expect the primary amine to act as an anchor based on the interactions described earlier. The IC$_{50}$ values of 0.6 and 1.5 nM were observed for 5 for PDGFRA-D842V and KIT-D816H, respectively (Fig. 3, Table 1). The data showed a 3–6-fold decrease in potency at the biochemical level compared to avapritinib. However, the reduction in cellular potency was substantially more pronounced and the

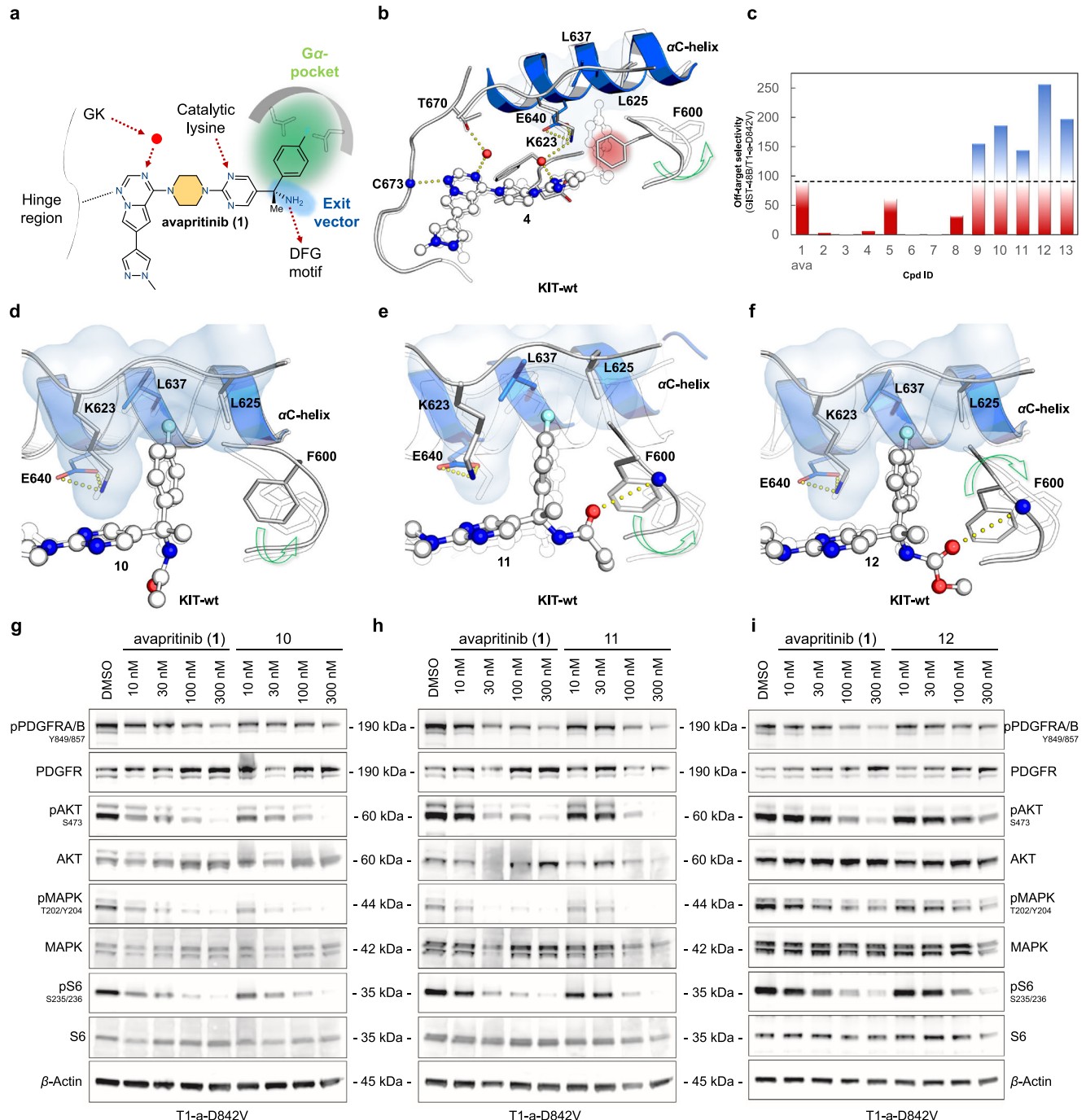

**Fig. 4 | Structural and biochemical analysis of the most active ligands synthesized during this project. a** Schematic representation of avapritinib's interaction pattern in PDGFRA and KIT. Shown are the main interactions with the hinge region, the catalytic lysine as well as the DFG motif. Further, a water-mediated interaction between ligand and protein (red dot) and the addressed Gα-pocket (green sphere) are indicated. **b** Comparison of complex crystal structures of **1** (PDB-ID: 8PQ9) and **4** (PDB-ID: 8PQA) bound to KIT-wt. Structures were aligned to their hinge regions. Without the fluorobenzene moiety (**4**), Phe600 of the Gly-rich loop is rotated inward the binding pocket so the Gα-pocket is covered by the phenyl side chain of the amino acid. **c** Graphical representation of the superior toxicity profiles of the synthesized ligands compared to avapritinib (**1**). Red: less selective on the control cell line than avapritinib (**1**). White: as selective on the control cell line as avapritinib (**1**). Blue: more selective on the control cell line than avapritinib (**1**). Ava: avapritinib (**1**). **d** Comparison of complex crystal structures of **1** (PDB-ID: 8PQ9) and

10 (PDB-ID: 8PQD) bound to KIT-wt, where Phe600 of the Gly-rich loop is slightly rotated toward the ligand binding pocket. **e** Comparison of complex crystal structures of **1** (PDB-ID: 8PQ9) and **11** (PDB-ID: 8PQE) bound to KIT-wt, where an additional interaction between ligand and Gly-rich loop can be observed. The regulatory αC-helix is slightly rotated upward. **f** Comparison of complex crystal structures of **1** (PDB-ID: 8PQ9) and **12** (PDB-ID: 8PQF) bound to KIT-wt, revealing an additional interaction between the carbamates oxygen and the backbone of Phe600 of the Gly-rich loop, which leads to a slight reorientation of the Gly-rich loop toward the ligand binding pocket. **g–i** Immunoblots to elucidate dose-dependent downregulation of pPDGFRA-D842V in T1-a-D842V cell lines. Cells were treated with DMSO (control) or inhibitors **1** (**g–i**), **10** (**g**), **11** (**h**) and **12** (**i**) for 24 h, lysed, blotted and incubated with PDGFRA and pPDGFRA/B specific antibodies as well as downstream protein-specific antibodies for evaluation of the inhibition ($n = 1$ biologically independent experiment).

compound showed no effect on secondary and tertiary mutations (Table 1, Supplementary Table 1). Therefore, we assumed that this moiety is not involved in crucial protein-ligand interactions that would result in a significantly reduced potency but presumably is essential for cellular activity due to differences in solubility or membrane permeability. Besides, **5** can be referred to as a comparator for **6** and **7** since they differ only in the heterocyclic linker moiety. While compounds **6** and **7** incorporate different spiro motifs replacing the piperazine, it is worth noting that even though structural analysis did not reveal any apparent significant protein-ligand interaction for the piperazine, these derivatives were significantly less potent than **5** in the biochemical as well as in the cellular experiments (Fig. 3, Table 1). Of note, the decrease in potency is much less prominent for the diazaspiro nonane **7** compared to diazaspiro heptane **6**, for example, in the case of GIST-T1 (-17-fold) compared to T1-a-D842V (>500-fold). From the significant differences in potency of **6** and **7** compared to **5**, we concluded that the piperazine plays a central role in locking the compound conformation, thus enabling the hinge contact and simultaneous optimal binding to the Gα-pocket. This finding again emphasizes the importance of this interaction pattern.

### Amine derivatization retains potency, optimizes off-target toxicity, and potentially overcomes toxicity-associated brain permeability

In general, all derivatives with modifications of the primary amine demonstrated strong inhibitory activities (Fig. 3, Table 1). We subjected them to the established crystallization program to gain structural insights that explain and underline the SAR. To our delight, we were able to obtain co-crystal structures at high resolutions for each of the inhibitors **8**-**12** bound to KIT-wt (1.5–2.0 Å) and **9** bound to PDGFRA-T674I (2.6 Å) (Supplementary Fig. 5). While the biochemical evaluation still revealed $IC_{50}$ values in the sub-nanomolar range, blocking the ability for proton donation through H-bonding by double methylation (**8**) was accompanied by a slight loss of potency by 3–5-fold (Fig. 3, Table 1). A single modification by ethylation (**9**) did not affect the potency significantly since the $IC_{50}$ values were almost the same as for avapritinib. However, a reduction between 1.7- and 3.5-fold in potency was observed in the cellular experiments (Table 1). A comparison of the obtained crystal structures indicated that the ethyl residue displaced one water molecule in the complex with KIT-wt. Yet, there were no other significant alterations in the complexes of KIT-wt and PDGFRA-T674I bound to **9** when compared to the complex structures bound to avapritinib, respectively (Supplementary Figs. 7e–h and 8c, e, f). Lower inhibitory activity was observed in KIT cell lines for both alkylated derivatives **8** and **9** compared to PDGFRA cell lines (Fig. 3, Table 1). The acetylated derivative **10** demonstrated strong inhibitory potency with $IC_{50}$ values of <0.1 and 0.5 nM for PDGFRA-D842V and KIT-D816H, respectively (Fig. 3, Table 1). Also, in the cellular evaluation, **10** exhibited potent inhibition of AL-mutated variants of PDGFRA and KIT (Table 1). The co-crystal structure of **10** bound to KIT-wt indicated conformational flexibility of the acetyl moiety, and it was modeled in two different orientations in the electron density (Fig. 4d, Supplementary Figs. 7i–l and 8g, h). The crystal structures of **11** and **12** showed a H-bond between the introduced carbonyl oxygen of the respective amide and the backbone of the Gly-rich loop, leading to a slight reorientation of the Gly-rich loop (Fig. 4e, f, Supplementary Figs. 7m–p and 8i–l). The rearrangement of Phe600, which is required for efficient binding into the Gα-pocket, appears to be facilitated by the methyl ester of **12** through further fixation of the phenylalanine via an additional hydrophobic interaction (Fig. 4f). To further emphasize this, derivative **13** with a phenyl ring was synthesized. Compound **13** showed a strong biochemical inhibitory potency of about 0.2 nM toward PDGFRA-D842V. In the cellular assay (with respect to T1-a-D842V), it lost 3-fold potency compared to avapritinib; however, **13** is completely inactive with respect to the control cell lines,

indicating an optimized off-target toxicity profile (Table 1, Fig. 4c, Supplementary Table 2). All amidated derivatives showed potent inhibitory activity for mutant-KIT (GIST-T1 and T1-D816E) and T1-a-D842V cell lines in the low double-digit nM range while exhibiting significantly lower inhibitory activity toward the control cell lines compared to avapritinib. Overall, modifications on the primary amine appeared to be well tolerated in terms of potency for mutant-PDGFRA and -KIT. Furthermore, these derivatives displayed an optimized off-target toxicity profile and reduced effects in all tested control cell lines (Table 1, Fig. 4c, Supplementary Table 2).

Because these compounds exhibited promising biochemical and cellular inhibitory properties, we further evaluated them in western blot studies to determine their effects on the phosphorylation of PDGFRA and on downstream signaling pathways. In line with the results from CellTiter-Glo (CTG) assays, treatment of T1-a-D842V cells revealed potent inhibition of PDGFRA phosphorylation similar to that of the approved drug avapritinib (Fig. 4g–i). Since these inhibitors demonstrated encouraging results and modification at this position seems tolerable in terms of potency, these results might also offer a handle to optimize pharmacokinetic parameters to avoid penetration of the BBB. To assess the appropriate permeability, in silico calculations (central nervous system multiparameter optimization (CNS MPO) scores[23,24]) were performed for all derivatives using MarvinSketch 22.13.0 software (Supplementary Table 3). The calculations indicated that **12** and **13** appear to have a reduced BBB penetration compared to avapritinib, with CNS MPO scores of 2.60 and 2.14, respectively, instead of 3.52 for avapritinib. Moreover, an MDCKII-MDR1 permeability assay was conducted with avapritinib and derivatives **10**-**13** (Supplementary Table 3)[25]. For avapritinib, a $P_{app}(A \rightarrow B)$ of 12.5 was determined, which indeed points to high penetration of the BBB. In contrast to the CNS MPO score, **12** showed no improvement in terms of BBB penetration with a $P_{app}(A \rightarrow B)$ of 12.4. Similarly, no significant improvements were observed for **10** and **11** with a $P_{app}(A \rightarrow B)$ of 11.3 and 10.1, respectively. In contrast, an improvement was achieved for **13** with a $P_{app}(A \rightarrow B)$ of 3.5, suggesting a significant reduction in BBB penetration. Hence, the amide-containing derivatives were the most potent inhibitors within this series, showing similar biochemical and cellular efficacy compared to avapritinib while exhibiting an optimized off-target profile and potentially offering the possibility to overcome toxicity-associated brain permeability. Nevertheless, additional experiments are necessary to further optimize the molecules with respect to brain permeability and toxicity and in vivo experiments are the next crucial step to further evaluate these findings.

In summary, the modifications described above appear to be a promising starting point for further optimization concerning off-target toxicity, blood-brain barrier penetration, and overcoming acquired drug resistance.

## Discussion

Cancer patients suffering from PDGFRA-D842V-driven GIST have limited treatment options. Until now, the only approved drug is the TKI avapritinib, whose detailed binding mode has only been inferred from various chemoinformatic approaches. Here, we have elucidated the binding mode of avapritinib in wild-type and mutant PDGFRA and KIT via protein X-ray crystallography. This outcome not only provides detailed information on the inhibitor's binding mode, but also highlights the impact of resistance mutations that occur during cancer treatment.

Furthermore, we report the identification of a sub-pocket (Gα-pocket) located in the N-lobe of the kinase domain, which is surrounded by amino acids of key regulatory elements such as the Gly-rich loop and the αC-helix. The Gα-pocket has not been previously reported for PDGFRA and KIT or related kinases. According to our studies, targeting the Gα-pocket offers great potential

to impact both potency and selectivity positively and to overcome acquired resistance mutations and should be considered for the development of next-generation inhibitors.

We demonstrated that avapritinib binds KIT and PDGFRA in a type 1.5 inhibitor fashion rather than a type 1 inhibitor as previously described in the literature.

Based on the structural findings, we designed and synthesized inhibitors with high potency and optimized off-target selectivity. We are thus convinced that the findings described herein will not only guide the design and the development of next-generation inhibitors to overcome toxicity-associated brain permeability and the current landscape of resistance mutations in GIST but will also have an impact on other cancers where mutated receptor tyrosine kinases are oncogenic drivers.

# Methods

## Reagents and materials
All supplies for the KIT and PDGFRA HTRF assay kit were purchased from CisBio (Bagnols-sur-Ceže, France). Active enzymes were purchased from ProQinase (KIT-D816H (#1041-0000-1 (002))) and Invitrogen (PDGFRA-D842V (PV4203, #2343231B)). Small volume (25 μL fill volume) white round-bottom 384-well plates were obtained from Greiner Bio-One GmbH (Solingen, Germany).

## Activity-based assay
The biochemical half-maximal inhibitory concentrations (IC$_{50}$) were determined with the TK HTRF KinEASE assay (Cisbio) according to the manufacturer's instructions. Briefly: 5 μL Kinase solution and 2.5 μL inhibitor solution (8% DMSO in HTRF buffer) were incubated for 30 min before the reaction was started by the addition of 2.5 μL starting solution containing ATP and substrate peptide. ATP concentrations were set at their respective $K_M$ values (12 μM for KIT-D816H, 14 μM for PDGFRA-D842V). The following substrate concentrations were used: 1 μM for KIT-D816H and 775 nM for PDGFRA-D842V. After reaction completion (KIT-D816H: 30 min, PDGFRA-D842V: 15 min), 10 μL of stop solution was added. The FRET signal was measured with an EnVision plate reader (PerkinElmer, Waltham, MA, US) ($\lambda$ ex 620 nm/$\lambda$ em 665 nm). The quotient of both intensities was recorded at 8 different inhibitor concentrations and data fit to a Hill 4-parameter equation with Quattro software suite (Quattro Research GmbH, Martinsried, Germany). Each reaction was performed in duplicates, and at least three independent determinations of each IC$_{50}$ were made.

## Construct design of KIT-WT and T670I
For crystallization studies, codon-optimized DNA encoding residues 551–934 of human KIT (Uniprot-ID: P10721), including an N-terminal His$_6$-tag and a thrombin cleavage site, was cloned into a pCDFDuet-1 expression vector together with the coding sequence for the phosphatase YopH (Uniprot-ID: P15273, amino acid 164–468). The construct is based on the published crystal structure of the c-KIT kinase domain (PDB-ID: 6GQK), including several amino acid substitutions and deletion of the kinase insert loop within the C-terminal subdomain (amino acids 688–765) that is replaced by EFVPYKVAPEDLYKDFLT[17]. The T670I mutation was introduced using site-directed mutagenesis (NEB).

## Protein expression and purification of KIT-WT and T670I
For protein expression, *Escherichia coli* BL21 (DE3) cells transformed with the pCDFDuet-1 vector (see above) were grown to an optical density of OD$_{600}$ = 0.6 (37 °C, 120 rpm), induced with 0.4 mM IPTG and incubated for 20 h (18 °C, 120 rpm). After expression, cells were harvested (4 °C, 5000 × $g$, 20 min), resuspended in lysis buffer (40 mM HEPES pH 8, 300 mM NaCl, 20 mM Imidazol, 10% (v/v) Glycerol, 1 mM TCEP, protease inhibitor cocktail complete EDTA-free (Roche)), lysed and incubated with 1% (w/v) CHAPS (slow stirring, 4 °C, 30 min)

followed by centrifugation (4 °C, 75,000 × $g$, 1 h). The supernatant was loaded onto a Nickel-affinity column. The protein was eluted with 40 mM HEPES pH 8, 300 mM NaCl, 250 mM Imidazol, 10% (v/v) Glycerol, 1 mM TCEP. For removal of the imidazol, the solution was dialyzed (40 mM HEPES, pH 8, 300 mM NaCl, 10 mM Imidazol, 10% (v/v) Glycerol, 0.5 mM TCEP) and simultaneously incubated with thrombin (1U/100 μg target protein) to remove the His$_6$-tag, followed by a second nickel-affinity chromatography collecting the flow through. For final purification, the protein was loaded onto a size exclusion chromatography (20 mM Tris pH 8, 200 mM NaCl, 1 mM TCEP) and concentrated to 6.6 mg/mL. The mass of the protein was confirmed by ESI-MS analysis.

## Crystallization experiments of KIT-wt and -T670I
The purified proteins were subjected to vapor diffusion crystallization experiments using the hanging drop method. Before crystallization, the proteins were incubated with a 3-fold excess of the corresponding ligands for 1 h at 4 °C. The crystallization conditions for KIT-wt and -T670I can be found in Supplementary Table 3.

## Construct design of PDGFRA WT and T674I
For crystallization studies, codon-optimized DNA encoding residues 550–973 of human PDGFRA (Uniprot-ID: P16234), including an N-terminal His$_{10}$-tag and a PreScission cleavage site, was cloned into a pIEX/Bac-3 expression vector. The construct is based on the published crystal structure of the PDGFRA kinase domain (PDB-ID: 5GRN), including several amino acid substitutions and deletion of the kinase insert loop within the C-terminal subdomain (amino acids 697–768). The T674I mutation was introduced using site-directed mutagenesis (NEB).

## Protein expression and purification of PDGFRA-wt and T674I
Transfection, virus generation, amplification and protein expression were carried out in *S. frugiperda* (SF9) cells (Thermo Fisher Scientific). After expression, cells were harvested (4 °C, 3000 × $g$, 20 min), resuspended in lysis buffer (20 mM Tris pH 8, 150 mM NaCl, 5 mM KCl, 20 mM Imidazol, 1 mM TCEP, protease inhibitor cocktail complete EDTA-free (Roche)), lysed and incubated with 1% (w/v) CHAPS (slow stirring, 4 °C, 30 min) followed by centrifugation (4 °C, 75,000×$g$, 1 h). The supernatant was loaded onto a Nickel-affinity column. The protein was eluted with 20 mM Tris pH 8, 150 mM NaCl, 5 mM KCl, 250 mM Imidazol, 1 mM TCEP. For removal of the imidazol, the solution was dialyzed (20 mM Tris pH 8, 50 mM NaCl, 1% (v/v) Glycerol, 1 mM TCEP) and simultaneously incubated with PreScission protease (ration protein 6:1 protease) to remove the His$_{10}$-tag, followed by a second nickel-affinity chromatography collecting the flow through. For final purification, the protein was loaded onto a size exclusion chromatography (20 mM Tris pH 8, 150 mM NaCl, 1 mM TCEP, 1% (v/v) Glycerol) and concentrated to 10 mg/mL. The mass of the protein was confirmed by ESI-MS analysis.

## Crystallization experiments of PDGFRA-wt and -T674I
The purified proteins were subjected to crystallization experiments using the hanging drop method. Before crystallization, the proteins were incubated with a 3-fold excess of the corresponding ligands for 1 h at 4 °C. The crystallization conditions for PDGFRA-wt and -T674I can be found in Supplementary Table 4.

## MDCKII-MDR1 assay
MDCKII-MDR1 cell line was licensed from the Netherlands Cancer Institute, Amsterdam, Netherlands, and internally expanded (Master and Working Banks). For assay runs, working banks are thawed and maximally passed until passage 20. The 96-well format MDCKII-MDR1 Assay is a routinely run assay at the LDC. In every assay run, three reference compounds are routinely tested (propranolol = highly

permeable, atenolol = poorly permeable, digoxin = efflux). To measure cellular permeability, MDCKII-MDR1 cells (the Netherlands Cancer Institute, Amsterdam, Netherlands) were seeded on a transwell membrane in a 96-well format and grown for 5 days. On day 4, TEER values were measured to determine the monolayer integrity. On day 5, compounds were applied at a concentration of 10 μM in HBSS to either the apical (A) or basolateral (B) side of a MDCKII-MDR1 cell monolayer and incubated for 2 h at 37 °C (each side $n = 3$ wells). For each compound, a calibration curve was measured in duplicate. Compound concentrations on each side of the monolayer were determined by LC-MS/MS (UFLC XR system (Shimadzu) coupled to a Qtrap 5500 instrument (Sciex)). Mean concentration values of A and B were used for the calculation of the apparent permeability ($P_{app}$) A → B and B → A. Further, apical volume [μL], basolateral volume [μL], area [cm²], time [s] and dosing [μM] were used for the calculation. The apparent permeability was calculated in the apical to basolateral (A → B) and basolateral to apical (B → A) directions according to the following equation: $P_{app}$ (A → B) = (ΔCB * VB * 0.001) / (Δt * A * Ct0, A). Lucifer yellow paracellular permeability assay was performed after sample collection to determine monolayer integrity.

### In silico CNS-MPO score calculations
MarvinSketch was used to draw, display and characterize chemical structures in terms of their physicochemical parameters and CNS-MPO scores, Marvin 22.13.0, Chemaxon (https://www.chemaxon.com).

### Generation of pharmacophore models
To evaluate more interactions of the crystallized ligands in the target proteins, LigandScout (v. 4.4.8) was used to generate two-dimensional pharmacophore models of each ligand bound to KIT or PDGFRA.

### Cell lines for GR$_{50}$ determinations
Imatinib (IM)-sensitive (GIST-T1, RRID: CVCL_4976) and IM-resistant (GIST-T1-D816E (RRID: CVCL_A9N0), GIST-T1-T670I (RRID: CVCL_A9M9), T1-V654A, T1-a-D842V, T1-a-D842V/G680R, T1-a-D842V/T674I/R, T1-a-5258 and GIST-48B (RRID: CVCL_M441)) cell lines were studied. GIST-T1 was established from human, untreated, metastatic GISTs, and carries a primary-activating mutation in exon 11 (V560_Y578del). GIST-48B, despite retaining the activating KIT mutation in all cells, expresses KIT transcript and protein at essentially undetectable levels. GIST-T1 was established by Takahiro Taguchi (Kochi University, Kochi, Japan). Additional information about GIST-T1-derived cell lines can be found in Supplementary Table 12. Cell lines were cultured in IMDM containing 10% FBS and 1% Penicillin/Streptomycin. SK-LMS-1 cell line was cultured in RPMI1640 containing 10% FBS and 1% Penicillin/Streptomycin. Cells were incubated at 37 °C and 5% CO₂. Cell lines are regularly authenticated by sequencing endogenous mutations in KIT, confirmation of KIT expression, and response to KIT inhibitor treatment. In the course of this study, all cell lines were regularly tested for mycoplasma contamination by PCR and by MycoAlert Mycoplasma Detection Kit (Lonza). KIT-negative cell lines serve as a negative control for the evaluation of KIT inhibitors.

### Cell lines for EC$_{50}$ determinations regarding off-target toxicity
Breast cancer cell lines ZR-75-1 (RRID: CVCL_0588, purchased from Sigma-Aldrich/ECACC) and MDA-MB-175VII (RRID: CVCL_1400, purchased from LGC Standards/ATCC) were studied to confirm off-target toxicity of avapritinib derivatives. ZR-75-1 cell line was cultured in RPMI-1640 medium (Gibco) supplemented with 10% FBS (PAN-Biotech) and 1% Penicillin/Streptomycin (Gibco). MDA-MB-175VII cells were cultured in DMEM medium (Gibco) supplemented with 10% FBS (PAN-Biotech) and 1% Penicillin/Streptomycin (Gibco). Cell line authenticity was confirmed by STR analysis at Eurofins Genomics. All cell lines were tested regularly for mycoplasma contamination using the Mycoplasmacheck Service at Eurofins Genomics. On day 0, cells were plated into white 384-well cell culture plates (Greiner Bio-One) using a Multidrop™ reagent dispenser (Thermo) at cell numbers that ensure linear and optimal luminescent signal intensity (ZR-75-1: 400 cells/well; MDA-MB-175VII: 800 cells/well). Following incubation for 24 h in a humidified atmosphere at 37 °C and 5% CO₂, cells were treated with inhibitors in serial dilutions ranging from 30 μM down to 0.1 nM using an Echo650 acoustic liquid handler (Beckman Coulter). Cell viability was analyzed on day 5 using the CellTiter-Glo® assay (Promega) using 500 ms integration time. The obtained data were normalized to the plate positive control (30 μM staurosporine) and negative control (DMSO) and subsequently analyzed and fitted with the Quattro Software Suite (Quattro Research) using a four-parameter logistic model. As quality control, the Z'-factor was calculated from 16 positive and negative control values. Only assay results showing a Z'-factor ≥0.5 were used for further analysis. All experimental points were measured in duplicates for each plate and were independently replicated in at least three plates.

### Reagents and antibodies
All primary and secondary antibodies used in this study were purchased from Cell Signaling Technologies (anti-tErk1/2 (order. no. 9102, 1:1000), anti-pERK1/2(Thr202/Tyr204) (order no. 9101, 1:1000), anti-tAKT (order no. 9272, 1:1000), anti-pAKT(Ser473) (order no. 9271, 1:1000), anti-S6 (order no. 2217, 1:1000), anti-pS6 (Ser235/236) (order no. 2211, 1:1000), anti-tPDGFRA (order no. 3174, 1:1000), anti-pPDGFRA (Tyr849)/PDGFRB (Tyr857) (order no. 3170, 1:1000), anti-beta-Actin (order no. 3700, 1:1000), secondary antibody anti-rabbit IgG, HRP-linked antibody (CST, order no. 7074, 1:1000), secondary antibody anti-mouse IgG, HRP-linked antibody (CST, order no. 7076, 1:1000)).

### Western blot analysis
Cells were plated in 6-well plates and, on the next day, treated with different inhibitors or vehicle control. After 24 h of treatment, lysis buffer (1% NP-40, 50 mmol/L Tris-HCl pH 8.0, 100 mmol/L sodium fluoride, 30 mmol/L sodium pyrophosphate, 2 mmol/L sodium molybdate, 5 mmol/L EDTA and 2 mmol/L sodium vanadate; freshly adding 0.1% 10 mg/mL aprotinin and leupeptin as well as 1% 100 mmol/L PMSF and 200 mmol/L sodium vanadate) was added, and cells were scraped off and then lysed while rotating for 1 h at 4 °C. Lysates were centrifuged at 4 °C for 30 min at 18,000rcf and protein concentration was determined using the Bio-Rad Protein Assay (Bio-Rad Laboratories). Protein concentration was adjusted to 2 μg/μL (if not otherwise specified), SDS-loading buffer (0.5 M Tris-HCl pH 6.7, 10% SDS, 2.5% DTT, 50% glycerol, and 0.05% bromophenol blue) was added and lysates were incubated for 5 min at 95 °C. Equal amounts of protein (30 μg per lane, if not otherwise specified) were separated on SDS-PAGE Gels (NuPAGE 4%–12%; Life Technologies) and blotted onto nitrocellulose-membranes (GE Healthcare/Amersham-Biosciences). After blocking with Net-G buffer (1.5 M NaCl, 50 mmol/L EDTA, 500 mmol/L Tris, 0.5% Tween 20, and 0.4% gelatine), membranes were incubated at 4 °C overnight with the respective primary antibody. After washing (Net-G), membranes were incubated for 2 h at room temperature with a secondary antibody (in Net-G) and washed again. Changes in protein expression and phosphorylation as visualized by chemiluminescence were captured and quantified using a FUJI LAS3000 system with Science Lab 2001 ImageGauge 4.0 software (Fujifilm Medial Systems). Usually, 2 to 4 gels/membranes were prepared from the same experiment/lysates to enable clean stains of proteins with similar or nearby molecular weight as well as stains of total proteins and their phosphorylated counterparts. Membranes were consecutively stained with different antibodies of different molecular weights. β-Actin served as loading control for each membrane, and a representative stain is shown.

**Reporting summary**

Further information on research design is available in the Nature Portfolio Reporting Summary linked to this article.

## Data availability

The data supporting the findings of this study are available in the paper and its Supplementary Information. The crystal structure data generated in this study have been deposited in the PDB database under accession codes 8PQ9, 8PQA, 8PQB, 8PQC, 8PQD, 8PQE, 8PQF, 8PQG, 8PQH, 8PQI, 8PQJ, 8PQK. The diffraction data is available for corresponding PDB-IDs at: https://www.proteindiffraction.org/project/8PQ9/, https://www.proteindiffraction.org/project/8PQA/, https://www.proteindiffraction.org/project/8PQB/, https://www.proteindiffraction.org/project/8PQC/, https://www.proteindiffraction.org/project/8PQD/, https://www.proteindiffraction.org/project/8PQE/, https://www.proteindiffraction.org/project/8PQF/, https://www.proteindiffraction.org/project/8PQG/, https://www.proteindiffraction.org/project/8PQH/, https://www.proteindiffraction.org/project/8PQI/, https://www.proteindiffraction.org/project/8PQJ/, https://www.proteindiffraction.org/project/8PQK/. Previously reported structures that were used in this publication are deposited under the following accession codes: 1T46, 6GQK, 6GQM, 6GQL, 1PKG, 3G0E, 4U0I, 6MOB, 7KHK, 7KHJ and 5GRN. Source data are provided with this paper.

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

## Acknowledgements

This work was co-funded by the German Research Foundation (DFG; BA 5214/1-2 (S.B.)|RA 1055/3-2 (D.R.)), the State of North Rhine-Westphalia (NRW), the European Union (European Regional Development Fund: Investing In Your Future) (EFRE-800400), DDHD (Drug Discovery Hub Dortmund, (D.R.)), the German Federal Ministry of Education and Research (InCa (01ZX2201B, (D.R.))), the Mercator Research Center Ruhr (MERCUR), IGNITE (Ex-2021-0033, (D.R. and S.B.)) and was supported by the "Netzwerke 2021" program, an initiative of the Ministry of Culture and Science of the State of North Rhine-Westphalia (CANcer TARgeting, NW21-062C, (D.R. and S.B.)). This work was supported by the Swiss Light Source of the Paul Scherrer Institute (SLS, Villingen, Switzerland) and The European Synchrotron Radiation Facility (ESRF, Grenoble, France, proposal MX-2391, https://doi.org/10.15151/ESRF-ES-744176088 and https://doi.org/10.15151/ESRF-ES-925653639, (D.R. and M.P.M.)). We thank Andreas Arndt for performing the biochemical assays.

## Author contributions

D.R. is responsible for initiating and supervising the project. T.S. designed and, together with R.G., synthesized the compounds. A.T. set up the biochemical assays. A.T. performed the protein production, co-crystallization experiments and data processing, and together with M.P.M., J.N. and S.B.K. performed the structure building. T.S. and A.T. conducted the structural analysis. T.M. generated the CRISPR/Cas9 cell lines. T.M., B.S.F., J.W. and S.S. performed the cell biology studies.

M.-L.Z. carried out MDCKII-MDR1 permeability assays. M.P.M., J.L. and S.B. advised the study. The manuscript was written through the contributions of all authors. All authors have given approval to the final version of the manuscript.

## Funding

## Competing interests

D.R. reports lecture and consulting fees from Pfizer, AZ, Sanofi, BI, and Bayer; is a shareholder of Centessa Pharmaceuticals plc., outside the submitted word. S.B. reports grants and personal fees from Blueprint Medicines, Incyte, and Novartis, PharmaMar, Eli Lilly & Co, Adcendo, Bayer, Blueprint Medicines, Boehringer Ingelheim, Cogent, Daiichi Sankyo, Deciphera, GSK, Exelixis, Novartis, Roche, PharmaMar. J.L. is a shareholder of Centessa Pharmaceuticals plc., outside the submitted work. The remaining authors declare no competing interests.
