## [Peer Review File · Nature Communications]

Reviewers' Comments:

Reviewer #1:

Remarks to the Author:

Teuber and colleagues obtained a number of interesting structures of the kinase domains of PDGFRA and KIT in complex with avapritinib. The impressive amount of generated data provides insights into the binding mode of avapritinib and resistance to this drug. This work unraveled a new pocket in the kinase domain.

Based on these results, the authors designed new compounds with interesting pharmacokinetics features, which could offer a possibility to overcome toxicity associated with brain penetration. Another goal was to improve the compound efficacy against resistant mutants. Supplementary table 1 suggests a modest advantage towards PDGFRA-T674R/KIT-T670I, but no real improvement towards double mutants such as PDGFRA-D842V-G680R. Other resistant mutations were not tested.

These results on the development of avapritinib derivatives are encouraging but rather preliminary. The authors should provide more compelling evidence that the new compounds provide an advantage in terms of brain penetration, toxicity and/or resistance.

Reviewer #2:

Remarks to the Author:

The manuscript submitted by Teuber and coworkers presents an excellent study of the interactions of avapritinib with wild type and gatekeeper-mutant forms of KIT and PDGFRA. The authors disclose several new crystal structures, including the first crystal structures of avapritinib with KIT-wt, KIT-T670I (first reported crystal of this KIT mutant), PDGFRA-T674I and other resolved crystal structures with structurally related avapritinib analogs. By the study of protein-ligand interactions, the authors identified a new sub-pocket (Ga-pocket) formed only upon ligand binding to the active site of KIT and PDGFRA. The role of this pocket in the kinase inhibition and the structural features that govern the avapritinib inhibitory activity, including avapritinib resistant mutation rationale, were assessed by biochemical, cellular and X-ray characterization using a small library of avapritinib analogs. Finally, the authors explore the functionalization of the primary amine group of avapritinib as a strategy to maintain the interaction with the Ga-pocket and modulate the pharmacokinetic properties for next-generation inhibitors to both overcome resistance mutations and reduce BBB penetration to reduce the side effects of avapritinib.

This is an interesting work that will could have a significant impact in the field of kinase inhibitors and drug discovery. In our opinion, the manuscript may be potentially suitable for Nature Communications, but there are important concerns that need to be addressed before this can be considered.

Main points:

- The intro is focused on the clinical importance of the activation loop-mutated forms of PDGFRA and KIT (D842V and D816V, respectively), which are the mutated targets to which avapritinib was developed to address resistant to imatinib. However, the full body of crystallographic work is focused on the wildtype and gatekeeper-mutant forms of these kinases. What is the clinical relevance of the gatekeeper mutation? Also, unless new co-crystal structures are included, the manuscript study does not provide evidence of the mode of binding of avapritinib to PDGFRA-D842V. Later, however, the biochemical studies are done against PDGFRA-D842V. These incongruences need to be addressed in a clear manner for the reader to understand the limitations of the study.

- The authors claim that avapritinib binds to the active conformation of PDGFRA and KIT wildtype and gatekeeper mutants, and creates a new pocket that does not exist in this conformation. However, they also state that this pocket exist in the inactive conformation, but it is occupied by the Leu839 of the activation loop. Therefore, by mimicking a key feature of the inactive conformation of the kinase, is not avapritinib forcing a "de facto" inactive conformation (even if it is DFG-in)? In fact, as shown in Supp Fig 2, panel "a" shows that the C-helix is displaced by avapritinib in a similar way as Leu839 does when the kinase adopt a DGF-out conformation. If

PDGFRA-D842V favours the active DFG-in conformation of the kinase, could avapritinib be taking the role of the mutated activation loop in creating the G-alpha pocket and thereby displace the C-helix into an essentially inactive conformation? It can be argued that avapritinib is a Type 1.5 kinase inhibitor that induces an inactive DFG-in C-helix-out conformation.

- In the manuscript the authors claim to have "identified potential avenues for further improvement, including means to obtain a better profile in terms of resistance mutations and possible ways to avoid adverse effects due to penetration of the blood-brain barrier". However, none of the novel avapritinib analogs presented in this work outperform avapritinib in single mutated KIT/PDGFR A nor double mutated KIT/PDGFR A. Therefore, the above claim is not substantiated with data, especially regarding BBB penetration. The authors show that conversion of the primary amine group into amides is tolerated for activity. However, based on in silico and cell permeability data, the reduction of BBB penetration for these derivatives is minimal. In fact, conversion of an amino group into amide is not expected nor a classical strategy to reduce BBB. One could argue that it could even increase BBB penetration.

- With respect to the BBB reduction strategy, as the authors also indicated, exploration of the solvent-exposed methyl-pyrazole moiety could serve to shed light on how to modulate BBB permeability in avapritinib's analogs. Additional medicinal chemistry work would be required to elucidate the role of this moiety in BBB permeability.

- Further characterization of the Ga-pocket would be helpful for future drug development efforts. The authors employed a "deconstruction" approach to investigate the interaction of the fluorobenzene group with the Ga-pocket. However, it is not clear what are the minimal structural features of avapritinib required to induce the formation of the Ga-pocket.

Additional comments to address by the authors:

- 1) Page 5, lines 105 – 110. The described interactions between 1 and Leu599, Leu825, Ala625, Tyr676, Val607 and Cys835 of PDGFRA-T674I are not shown in Supp. Fig. 1a.
- 2) Page 6, lines 120 – 123. Please, clarify if the Ga-pocket is observed only with avapritinib and if it does not appear with other reported crystallized inhibitors.
- 3) Page 7, line 144. Replace "(Fig. 2a, b)" by "(Fig. 2a)".
- 4) Page 7, line 146. Replace "(Fig. 2a, b)" by "(Fig. 2b)".
- 5) Page 8, lines 173 – 174. The mentioned water-mediated interactions in KIT with Lys723 and Asp810 are not properly represented in Supp. Fig. 1b. Asp810 is not labelled and there is not any interaction shown with the primary amine of avapritinib.
- 6) Page 9, lines 185 – 187. The reasons behind the use of heterocyclic spiro-based linkers should be clearly described.
- 7) Page 9, lines 187 – 190. What have you based on to identify those two points of modification to reduce BBB permeability? What is the "knowledge gained"?
- 8) Page 9, lines 192 – 193. In the synthetic procedure piperazine was not used directly to prepare compound 3. Instead, 1-Boc-piperazine was used followed by Boc deprotection.
- 9) Page 9, line 201. It is stated that thirteen novel compounds were synthesized. However, the 4-chloro-6-(1-methyl-1H-pyrazol-4-yl)pyrrolo[2,1-f][1,2,4]triazine is commercially available and compound 5 has been previously described for other applications (WO2015/057873A1, page 21, compound 7).
- 10) Page 10, line 208. Replace "GIST48B" for "GIST-48B".
- 11) Page 10, lines 222 – 223. "Overall, the data indicated that the interaction of the triazine with the hinge region via H-bonding is less important than might be expected". This conclusion cannot be reached from this study, as all the synthesized analogs maintain the key structural elements to establish the interactions with the hinge.
- 12) Page 11, line 242. After "biochemical level" include "compared to avapritinib".
- 13) Page 12, line 254. Change "5" for "5".
- 14) Page 12, lines 254 – 257. "we concluded that the piperazine plays a central role in increasing the solubility". There is not sufficient data to support this conclusion. The physicochemical properties of 5, 6 and 7 are very similar and no solubility studies have been performed. These three compounds differ in the "linker" length, which influences how the rest of the molecule interacts with the target, and therefore affecting the affinity.

- 15) Page 12, line 269. The term "2-fold" is not accurate. Replace it for "between 1.7 and 3.5-fold".
- 16) Page 19, Fig. 1.
- a. Fig. 1 a) and b). To help understand how the presented inhibitors in b) bind to KIT, color-code the different parts of the molecules presented in b) so they match the surface colors in a). Also, indicate in the figure caption which protein is represented in the figure.
- b. Fig. 1 d) and e). The surface of avapritinib is colored green in d) as well as JMD and the Ga-pocket is colored blue, but the Ga-pocket is green in e) and also c) and f), which makes the figure difficult to understand. Please change the colors so each color corresponds to only one element. Keep this color consistent through the entire manuscript.
- c. In figures c) and d) include the terms "Inactive" an "Active", correspondingly, to indicate the state of the kinase.
- 17) Page 20, Fig. 2. To help understand the figures better, please include in panels a), b), c) and d) the name of the protein displayed and the compound bound. The same for Fig. 3, Supp. Fig. 1, Supp. Fig. 2 and Supp. Fig. 5.
- 18) Page 21, Fig. 3 e). Include the stick representation for F600.
- 19) Page 22, Table 1. The title indicates "IC50 and EC50 determinations". However, in the table it also appears GR50. Does GR state for growth rate inhibition? There is no explanation in the text regarding why both EC50 and GR50 have been used. In table 1 also appears "CTG", this abbreviation has not been used in the manuscript. Does it refer to CellTiter-Glo assays? If so, include it in the table description and in the manuscript (line 299). In the HTRF IC50 column, replace "PDGFR-D842V" by "PDGFRA-D842V"
- 20) Supp. Fig. 3. In the figure caption, please indicate the molecule presented in the figure (avapritinib?).
- 21) Supp. Tab. 1. The title indicates "EC50 determinations" but what is shown are GR50. Indicates the meaning of CTG.
- 22) Additional supp. material is required:
- a. Include a measurement of purity (HPLC analysis) for all the final compounds synthesized and tested.
- b. Include 1H-NMR and 13C-NMR spectra of all final molecules synthesized and tested.

Reviewer #3:

Remarks to the Author:

The authors provide an impressive experimental tour de force to understand the structural basis of activity of the kinase inhibitor avapritinib and its structure-based optimization. The extend of the structural and medicinal chemistry work is impressive. The experimental work is conducted well and supports the claims.

The work is presented clearly and the manuscript is concise.

A few points would strengthen the study considerably.

- while the achievements are very impressive, it may be useful to emphasize the broader relevance of the work outside of a kinase medchem focused interest. For example GIST is fortunately a very rare cancer and the work presented here is motivated largely by addressing mainly the needs of what is a small sub-population of patients whose disease does not respond to imatinib, sunitinib or repretinib. It may be of interest to the general audience of Nat Comm to understand what the scope of the problem is.

- line 111: A ligplot-style illustration of the interactions would be helpful.

- line 118: While the authors define the Galpha pocket, it is confusing terminology with pockets in other kinases near the alphaG-helix and should be renamed. Otherwise it is the Galpha pocket which is near the P-loop (also referred to as the Gly-rich loop) and one of several alpha helices.

How does avapritinib binding affect the R-spine? Can the R-spine assemble without avapritinib? The figures are somewhat unclear about how the pocket forms. This should be clarified.

The details of the experimental methods are unacceptable. For example, the crystallization conditions for the protein complexes are missing.

Response to Decision Letter: Teuber, Schulz, *et al.*

Manuscript NCOMMS-23-30228

Reviewer: 1

Teuber and colleagues obtained a number of interesting structures of the kinase domains of PDGFRA and KIT in complex with avapritinib. The impressive amount of generated data provides insights into the binding mode of avapritinib and resistance to this drug. This work unraveled a new pocket in the kinase domain. Based on these results, the authors designed new compounds with interesting pharmacokinetics features, which could offer a possibility to overcome toxicity associated with brain penetration.

We are grateful and thank the reviewer for the positive assessment of our work.

These results on the development of avapritinib derivatives are encouraging but rather preliminary. The authors should provide more compelling evidence that the new compounds provide an advantage in terms of brain penetration, toxicity and/or resistance.

We would like to emphasize that our objective was not to create stronger avapritinib derivatives. Instead, the avapritinib derivatives we have produced and studied were designed to experimentally understand how this new drug binds to its target oncogenes, such as KIT and PDGFRA. This helps to identify the essential components of this drug for efficient and selective binding. We are confident that this knowledge will be crucial in developing future drugs that can counteract acquired drug resistance and prevent brain penetration. However, in response to the reviewer's concerns about drug resistance, we have further analyzed our derivatives using two additional clinically relevant cancer cell lines. These include a common secondary KIT mutation (T1-V654A) and a tertiary PDGFRA mutation (T1-a-5258) (see Supp. Tab. 1). Unfortunately, the tested derivatives did not show an improved inhibitory profile compared to avapritinib. With regard to toxicity, we performed further CTG assays in two non-KIT/PDGFRA-dependent breast cancer cell lines (ZR-75-1, CVCL_0588; MDA-MB-175VII, CVCL_1400) as well as in a kinase-independent leiomyosarcoma cell line (SK-LMS-1, CVCL_0628). There was no evidence of toxicity as the cell lines were even less sensitive to treatment compared to avapritinib (Supp. Tab. 2).

Furthermore, we would like to highlight that our compounds have already demonstrated a significant reduction in off-target toxicity compared to avapritinib (up to 7.5-fold) when assessed in the GIST-48B cell line, as shown in Tab. 1 and Fig. 3c. This particular cell line is not dependent on KIT or PDGFRA and is widely used in the GIST field to study off-target toxicity.¹ In addition, we performed MDCKII-MDR1 assays to validate additional compounds with respect to their brain permeability. The best derivative (13) showed a significant reduction in likelihood to cross the BBB ($P_{appA \rightarrow B}$: 3.5), further strengthening our hypothesis, that derivatization of this position could be a good handle to overcome BBB penetration.

Reviewer: 2

The manuscript submitted by Teuber and coworkers presents an excellent study of the interactions of avapritinib with wild type and gatekeeper-mutant forms of KIT and PDGFRA.

We sincerely appreciate and thank the reviewer for their favorable evaluation of our research.

The intro is focused on the clinical importance of the activation loop-mutated forms of PDGFRA and KIT (D842V and D816V, respectively), which are the mutated targets to which avapritinib was developed to address resistant to imatinib. However, the full body of crystallographic work is focused on the wildtype and gatekeeper-mutant forms of these kinases. What is the clinical relevance of the gatekeeper mutation? Also, unless new co-crystal structures are included, the manuscript study does not provide evidence of the mode of binding of avapritinib to PDGFRA-D842V. Later, however, the biochemical studies are done against PDGFRA-D842V. These incongruences need to be addressed in a clear manner for the reader to understand the limitations of the study.

This point is well taken. The gatekeeper mutation is one of the most challenging resistance mutations occurring in the majority of protein kinase targets, including but not limited to KIT and PDGFRA. From the large number of clinically relevant mutant variants of KIT and PDGFRA we have cloned, expressed and purified so far, only wild-type and gatekeeper mutant variants resulted in protein crystals suitable for X-ray diffraction experiments. We addressed this in the manuscript. It now reads:

“Against this background, attempts have been made to establish expression and crystallization conditions for several clinically relevant mutants of KIT and PDGFRA. However, to date, these attempts only resulted in successful expression of stable and active wild-type (wt) KIT and PDGFRA proteins as well as gatekeeper mutants. The currently available proteins have been selected as model systems and subjected to crystallization studies. The structures obtained provide valuable insights and also allow us to deduce the effects conferred by other resistance mutations.”

The authors claim that avapritinib binds to the active conformation of PDGFRA and KIT wildtype and gatekeeper mutants, and creates a new pocket that does not exist in this conformation. However, they also state that this pocket exist in the inactive conformation, but it is occupied by the Leu839 of the activation loop. Therefore, by mimicking a key feature of the inactive conformation of the kinase, is not avapritinib forcing a "de facto" inactive conformation (even if it is DFG-in)? In fact, as shown in Supp Fig 2, panel "a" shows that the C-helix is displaced by avapritinib in a similar way as Leu839 does when the kinase adopt a DGF-out conformation. If PDGFRA-D842V favours the active DFG-in conformation of the kinase, could avapritinib be taking the role of the mutated activation loop in creating the G-alpha pocket and thereby displace the C-helix into an essentially inactive conformation? It can be argued that avapritinib is a Type 1.5 kinase inhibitor that induces an inactive DFG-in C-helix-out conformation.

The reviewer makes an excellent point, and we agree that classification as a type 1.5 inhibitor is more precise to describe the binding characteristics of avapritinib. The manuscript now reads:

“(…) Avapritinib mimics this key feature, thereby displacing the AL and forcing an α C-helix-out yet DFG-in conformation (Supp. Fig. 3). (…) Thus, our detailed analysis showed that binding of avapritinib to a DFG-in, but α C-helix-out conformation characterizes it as a type 1.5 inhibitor rather than a canonical type 1 inhibitor. (…) These conformational findings, which only became evident with the analysis of the obtained crystal structures, will be of great importance for the design and development of next-generation inhibitors to overcome acquired drug resistance and it highlights the importance of high-resolution complex crystal structures in drug discovery.”

In the manuscript the authors claim to have "identified potential avenues for further improvement, including means to obtain a better profile in terms of resistance mutations and possible ways to avoid adverse effects due to penetration of the blood-brain barrier". However, none of the novel avapritinib analogs presented in this work outperform avapritinib in single mutated KIT/PDGFR α nor double mutated KIT/PDGFR α . Therefore, the above claim is not substantiated with data, especially regarding BBB penetration. The authors show that conversion of the primary amine group into amides is tolerated for activity. However, based on in silico and cell permeability data, the reduction of BBB penetration for these derivatives is minimal. In fact, conversion of an amino group into amide is not expected nor a classical strategy to reduce BBB. One could argue that it could even increase BBB penetration.

We are grateful for these comments. The conversion of the amino group to an amide provided us the opportunity to increase the molecular weight of the molecule and to vary its polar surface area; properties known to influence BBB penetration.^{2, 3} As highlighted above in response to reviewer #1, the primary objective of our work was to elucidate the binding mode of avapritinib and to gain a better understanding of key pharmacophoric features, as there is no information available in the literature to date. Nevertheless, we performed MDCKII-MDR1 permeability assays to evaluate additional compounds 12 and 13 to strengthen the point about reduced BBB penetration (Supp. Tab. 3). Regarding the resistance mutations, we added additional data for a secondary KIT mutation (V654A) and a tertiary PDGFR α mutation (T1-a-5258) (Supp. Tab. 1). Unfortunately, we did not observe improvement compared to avapritinib. With respect to off-target toxicity, we further added KIT and PDGFR α -independent cancer cell lines ZR-75-1, MDA-MB-175VII and SK-LMS-1 (Supp. Tab. 2). Although avapritinib shows low activity, our compounds are even less active than avapritinib on these cell lines, indicating an even better off-target profile.

With respect to the BBB reduction strategy, as the authors also indicated, exploration of the solvent-exposed methyl-pyrazole moiety could serve to shed light on how to modulate BBB permeability in avapritinib's analogs. Additional medicinal chemistry work would be required to elucidate the role of this moiety in BBB permeability.

We agree with the referee that additional medicinal chemistry work would be required to explore the solvent-exposed methyl-pyrazole as an additional exit vector to block BBB penetration of avapritinib. A recent patent search revealed that BluePrint Medicines has indeed pursued this strategy and developed BLU-263 (WO2022/081626), which seems to be less BBB permeable compared to avapritinib and is now in clinical trials for the treatment of systemic mastocytosis.

Further characterization of the G α -pocket would be helpful for future drug development efforts. The authors employed a "deconstruction" approach to investigate the interaction of the fluorobenzene group with the G α -pocket. However, it is not clear what are the minimal structural features of avapritinib required to induce the formation of the G α -pocket.

The reviewer raises an excellent point, and further characterization of the G α -pocket formation is indeed an ongoing research topic in our laboratory. Since we do not currently know whether the underlying mechanism of the formation of the G α -pocket is via induced fit or conformational selection, we envision sophisticated titration experiments via NMR and appropriate isotope labeling strategies to unravel this. Since the outcome of such experiments cannot be predicted and the significance for further clinical developments of corresponding derivatives is rather limited, we believe that the experiments are far beyond the scope and intents of this manuscript.

1) Page 5, lines 105– 110. The described interactions between 1 and Leu599, Leu825, Ala625, Tyr676, Val607 and Cys835 of PDGFRA-T674I are not shown in Supp. Fig. 1a.

We have corrected this in Supp. Fig. 1. We now also include 2D plots of the pharmacophore model of the inhibitors (Supp. Fig. 2, Supp. Fig. 8).

2) Page 6, lines 120 – 123. Please, clarify if the α -pocket is observed only with avapritinib and if it does not appear with other reported crystallized inhibitors.

“To our knowledge, the α -pocket has not been targeted by other reported inhibitors in the field of GIST and has only been observed upon the binding of avapritinib and corresponding derivatives.”

3) Page 7, line 144. Replace "(Fig. 2a, b)" by "(Fig. 2a)".

done

4) Page 7, line 146. Replace "(Fig. 2a, b)" by "(Fig. 2b)".

done

5) Page 8, lines 173 – 174. The mentioned water-mediated interactions in KIT with Lys723 and Asp810 are not properly represented in Supp. Fig. 1b. Asp810 is not labelled and there is not any interaction shown with the primary amine of avapritinib.

We thank the reviewer for pointing this out. It has been corrected and can now be seen in Supp. Fig. 1e.

6) Page 9, lines 185 – 187. The reasons behind the use of heterocyclic spiro-based linkers should be clearly described.

The text now reads:

“Second, instead of the piperazine, heterocyclic spiro-based linkers were investigated. This was done to test different linker lengths connecting the hinge binding element to the α pocket moiety, as well as to explore their chemical nature as complex, highly saturated structures with a significant three-dimensional character. (Fig. 3a, Supp. Fig. 6a).”

7) Page 9, lines 187 – 190. What have you based on to identify those two points of modification to reduce BBB permeability? What is the “knowledge gained”?

The manuscript now reads:

“The crystal structures obtained during this work revealed two suitable solvent-exposed exit vectors that would qualify for the modification of avapritinib to potentially block BBB penetration without negatively affecting potency”

8) Page 9, lines 192 – 193. In the synthetic procedure piperazine was not used directly to prepare compound 3. Instead, 1-Boc-piperazine was used followed by Boc deprotection.

Thank you for pointing this out! The manuscript now reads:

“For the deconstruction, nucleophilic aromatic substitutions were conducted based on the conversion of the commercially available 4-chloro-6-(1-methyl-1H-pyrazol-4-yl)pyrrolo[2,1-f][1,2,4]triazine 2 with either 1-Boc-piperazine, followed by acid-catalyzed Boc-deprotection, or a nucleophilic substitution with 2-(piperazin-1-yl)pyrimidine yielding compounds 3 and 4, respectively (Supp. Fig. 6a).”

9) Page 9, line 201. It is stated that thirteen novel compounds were synthesized. However, the 4-chloro-6-(1-methyl-1H-pyrazol-4-yl)pyrrolo[2,1-f][1,2,4]triazine is commercially available and compound 5 has been previously described for other applications (WO2015/057873A1, page 21, compound 7).

We have corrected this. The sentence now reads:

“The successfully synthesized avapritinib-based inhibitors as well as pyrrolo-triazine 2 were biochemically and cellularly evaluated against mutant forms of KIT and PDGFRA (Tab. 1, Supp. Tab. 1).”

10) Page 10, line 208. Replace "GIST48B" for "GIST-48B".

done

11) Page 10, lines 222 – 223. "Overall, the data indicated that the interaction of the triazine with the hinge region via H-bonding is less important than might be expected". This conclusion cannot be reached from this study, as all the synthesized analogs maintain the key structural elements to establish the interactions with the hinge.

We rephrased the sentence. It now reads:

“Overall, the data indicate that binding to the G α -pocket is an essential structural element for potent ligand binding.”

12) Page 11, line 242. After "biochemical level" include "compared to avapritinib".

done

13) Page 12, line 254. Change "5" for "5".

done

14) Page 12, lines 254 – 257. "we concluded that the piperazine plays a central role in increasing the solubility". There is not sufficient data to support this conclusion. The physicochemical properties of 5, 6 and 7 are very similar and no solubility studies have been performed. These three compounds differ in the "linker" length, which influences how the rest of the molecule interacts with the target, and therefore affecting the affinity.

We rephrased the sentence. It now reads:

“From the significant differences in potency of 6 and 7 compared to 5, we concluded that the piperazine plays a central role in locking the compound conformation, thus enabling the hinge contact and simultaneous optimal binding to the G α -pocket.”

15) Page 12, line 269. The term "2-fold" is not accurate. Replace it for "between 1.7 and 3.5-fold".

done

16) Page 19, Fig. 1. a. Fig. 1 a) and b). To help understand how the presented inhibitors in b) bind to KIT, color-code the different parts of the molecules presented in b) so they match the surface colors in a). Also, indicate in the figure caption which protein is represented in the figure. b. Fig. 1 d) and e). The surface of avapritinib is colored green in d) as well as JMD and the G α -pocket is colored blue, but the G α -pocket is green in e) and also c) and f), which makes the figure difficult to understand. Please change the colors so each color corresponds to only one element. Keep this color consistent through the entire manuscript. c. In figures c) and d) include the terms "Inactive" an "Active", correspondingly, to indicate the state of the kinase.

This is an excellent point. We updated the figures and figure legends accordingly.

17) Page 20, Fig. 2. To help understand the figures better, please include in panels a), b), c) and d) the name of the protein displayed and the compound bound. The same for Fig. 3, Supp. Fig. 1, Supp. Fig. 2 and Supp. Fig. 5.

This is an excellent advice. We updated the figures and figure legends accordingly.

18) Page 21, Fig. 3 e). Include the stick representation for F600.

done

19) Page 22, Table 1. The title indicates "IC₅₀ and EC₅₀ determinations". However, in the table it also appears GR₅₀. Does GR state for growth rate inhibition? There is no explanation in the text regarding why both EC₅₀ and GR₅₀ have been used. In table 1 also appears "CTG", this abbreviation has not been used in the manuscript. Does it refer to CellTiter-Glo assays? If so, include it in the table description and in the manuscript (line 299). In the HTRF IC₅₀ column, replace "PDGFR-D842V" by "PDGFRA-D842V"

Thank you for pointing this out to us. We revised the table description accordingly and now show GR₅₀ values throughout the manuscript and also included the explanation for GR₅₀ in the caption as well as the abbreviation for CTG in the manuscript. Also, we replaced "PDGFR-D842V" by "PDGFRA-D842V".

20) Supp. Fig. 3. In the figure caption, please indicate the molecule presented in the figure (avapritinib?).

Done as can be seen at point 17) and in the supplementary information (now Supp. Fig. 5).

21) Supp. Tab. 1. The title indicates "EC₅₀ determinations" but what is shown are GR₅₀. Indicates the meaning of CTG.

We corrected this.

22) Additional supp. material is required: a. Include a measurement of purity (HPLC analysis) for all the final compounds synthesized and tested. b. Include ¹H-NMR and ¹³C-NMR spectra of all final molecules synthesized and tested.

¹H- and ¹³C-NMR spectra, as well as HPLC analysis, are provided for each final compound in the supplementary information below the corresponding synthetic procedures.

Reviewer: 3

The authors provide an impressive experimental tour de force to understand the structural basis of activity of the kinase inhibitor avapritinib and its structure-based optimization. The extend of the structural and medicinal chemistry work is impressive. The experimental work is conducted well and supports the claims. The work is presented clearly and the manuscript is concise.

We value and express our gratitude to the reviewer for their positive evaluation of our study.

A few points would strengthen the study considerably. - while the achievements are very impressive, it may be useful to emphasize the broader relevance of the work outside of a kinase medchem focused interest. For example GIST is fortunately a very rare cancer and the work presented here is motivated largely by addressing mainly the needs of what is a small

sub-population of patients whose disease does not respond to imatinib, sunitinib or repretinib. It may be of interest to the general audience of Nat Comm to understand what the scope of the problem is.

The reviewer correctly points out that GIST is a comparatively rare cancer. Nevertheless, GIST is the prime example of precision oncology, where patients benefit from drugs specifically tailored to target the genetic abnormalities of protein kinases inherent in the disease. The drug in question, avapritinib, was designed to inhibit the PDGFRA mutation variant D842V found in GIST patients. This drug's FDA approval in 2020, backed by fast track, breakthrough therapy, and orphan drug designations, attests to its critical role in GIST treatment. Our study of avapritinib's interaction with the kinase domains of KIT and PDGFRA, understanding its structural features relevant to the unwanted BBB penetration, and knowledge about its potential weak points against drug resistance, is vital. These insights guide not only the design and development of next-generation drugs for KIT and PDGFRA-driven GIST, but other cancers where mutated receptor tyrosine kinases are the oncogenic drivers, such as EGFR and HER2 in non-small cell lung cancer and acquired drug resistance to approved drugs is a challenge.

To highlight the importance of GIST as paramount for precision oncology, we added the following sentence to the introduction:

“Therefore, GIST is a beacon of precision oncology, demonstrating the transformative impact of personalized treatments for unique genetic abnormalities in cancer therapy.”

To follow up on this, we also added the following sentence in the discussion:

“We are thus convinced that the findings described herein will not only guide the design and the development of next-generation inhibitors to overcome toxicity-associated brain permeability and the current landscape of resistance mutations in GIST, but will also have an impact on other cancers where mutated receptor tyrosine kinases are oncogenic drivers.”

line 111: A ligplot-style illustration of the interactions would be helpful.

This point is well taken. We now include 2D interaction analysis from LigandScout as Supp. Fig. 2 and Supp. Fig. 8 for all complex crystal structures discussed in the manuscript.

line 118: While the authors define the Galpha pocket, it is confusing terminology with pockets in other kinases near the alphaG-helix and should be renamed. Otherwise it is the Galpha pocket which is near the P-loop (also referred to as the Gly-rich loop) and one of several alpha helices.

We are sorry that the reviewer finds the term $G\alpha$ -pocket potentially misleading. Unfortunately, to the best of our knowledge, there is no general convention for naming binding pockets. Within structural biology, the P-loop is often used in the small GTPase field, while the glycine-rich loop is more commonly used in the kinase field. Therefore, we decided to stick with the term $G\alpha$ -pocket to best describe the location of the newly discovered binding pocket in KIT and PDGFRA.

How does avapritinib binding affect the R-spine? Can the R-spine assemble without avapritinib? The figures are somewhat unclear about how the pocket forms. This should be clarified.

We thank the reviewer for bringing up this excellent question about the R-spine. We analyzed different states of KIT (ATP-bound, (PDB-ID: 1PKG)), ligand-bound DFG-in (avapritinib (PDB-ID: 8PQ9)), and ligand-bound DFG-out (imatinib (PDB-ID: 1T46)) as well as an PDGFRA inactive apo structure (PDB-ID: 8PQJ). We conclude that the R-spine can assemble in the active kinase conformations as seen in the ATP-bound, public structure, which is, as expected, not the case for the inactive conformations. Compared to the avapritinib-bound structure, only minor differences were observed. Therefore, we would argue that avapritinib binding itself does not have a huge impact on the formation of the R-spine. To further emphasize this, we created an additional figure for the supplementary information (Supp. Fig. 4). In the manuscript it reads as follows:

“(…) Following these findings, we analyzed different states of KIT and PDGFRA, including ligand-bound and apo-crystal structures, with a particular focus on analysis of the assembly of the R-spine (Supp. Fig. 4). In the inactive DFG-out conformations (apo, PDB ID: 8PQJ, type II inhibitor bound, PDB ID: 1T46), an absence of R-spine formation was observed, as expected (Supp. Fig. 4d, e). Conversely, in the activated ATP-bound state (PDB-ID: 1PKG), R-spine assembly was observed (Supp. Fig. 4c). Interestingly, the co-crystal structures of avapritinib revealed that the R-spine is also able to assemble (Supp. Fig. 4a, b).”

As we have stated in our response to reviewer #2, the analysis of the mechanisms (induced fit vs. conformational selection) by which the α -pocket is formed is currently investigated by NMR but far beyond the scope of this manuscript.

The details of the experimental methods are unacceptable. For example, the crystallization conditions for the protein complexes are missing.

We apologize for this! The full corresponding information was added to the PDB during submission but was indeed missing from the manuscript. We have now added a detailed table of crystallization conditions for each structure described in the manuscript in the Supplementary Information. We have also added a table to clarify the nomenclature of the cell lines and CTG measurements used. As requested by reviewer #2, we have also included ^1H and ^{13}C NMR spectra and HPLC spectra for the final inhibitors in the Supplement.

In addition to the above, other minor corrections have been made and are marked in yellow throughout the manuscript and in the Supplementary Information. We also performed additional assays and updated the values in the tables (marked in yellow). No discrepancies were observed.

References

1. Mühlenberg T, *et al.* Inhibitors of deacetylases suppress oncogenic KIT signaling, acetylate HSP90, and induce apoptosis in gastrointestinal stromal tumors. *Cancer Res* **69**, 6941-6950 (2009).
2. Pardridge WM. Drug transport across the blood-brain barrier. *J Cereb Blood Flow Metab* **32**, 1959-1972 (2012).
3. Kadry H, Noorani B, Cucullo L. A blood-brain barrier overview on structure, function, impairment, and biomarkers of integrity. *Fluids Barriers CNS* **17**, 69 (2020).

Reviewers' Comments:

Reviewer #1:

Remarks to the Author:

The authors have clarified a number of key issues and have much improved their manuscript, which is now of general interest to the scientific community.

Comments:

- I am not convinced by the "1.5" terminology. Overall, avapritinib is a type I inhibitor based on its activity on AL mutants and on the positions of both AL and R-spine, even though there is a deviation in the alpha-helix position.

- The experiments performed to model BBB penetration remain preliminary, in the absence of in vivo data. Off-target toxicity is even more difficult to estimate using model systems. The authors should at least state these points and, in general, discuss the limitations of their study.

Reviewer #2:

Remarks to the Author:

Excellent revision. The authors have addressed all our concerns.

Reviewer #3:

Remarks to the Author:

The reviewers have addressed my concerns from the previous round of review and the manuscript has substantially improved.

Reviewer #4:

None

Response to Decision Letter: Teuber, Schulz, et al.

Manuscript NCOMMS-23-30228

Reviewer: 1

The authors have clarified a number of key issues and have much improved their manuscript, which is now of general interest to the scientific community.

We are grateful for the positive assessment of our revisions.

Comments:

- I am not convinced by the "1.5" terminology. Overall, avapritinib is a type I inhibitor based on its activity on AL mutants and on the positions of both AL and R-spine, even though there is a deviation in the alpha-helix position.

We acknowledge the reviewer's perspective on avapritinib as a potent inhibitor predominantly active against AL mutants and engaging the DFG-in conformation, typically characteristic of type I inhibitors. However, we would like to draw attention to the comprehensive definition provided by ROSKOSKI on type 1.5 inhibitors, which underscores their unique binding dynamics. According to this definition, a type 1.5 inhibitor is characterized by its interaction within and around the ATP-binding pocket of an enzyme in an inactive DFG-Asp in state.¹ Our observations reveal that while avapritinib does indeed allow the formation of the regulatory spine (R-spine), indicative of an active conformation, it simultaneously impacts other critical structural elements, such as the α C-helix and the Gly-rich loop, stabilizing an inactive conformation. This dual influence is pivotal for the formation of the $G\alpha$ -pocket, a feature integral to the inactive kinase conformation. Furthermore, the structural analysis of avapritinib shows a clear mimicry of features inherent to the inactive kinase conformation, in particular the occupation of the $G\alpha$ -pocket by the AL (e.g. PDGFRA, Leu839; as detailed in Supplementary Figure 3c). Therefore, we respectfully assert that the subtleties in the conformational engagement of avapritinib with its target kinase, encompassing elements from both active and inactive states, justify its classification as a type 1.5 inhibitor. This nuanced understanding is crucial for accurately categorizing avapritinib and comprehending its mechanism of action.

- The experiments performed to model BBB penetration remain preliminary, in the absence of in vivo data. Off-target toxicity is even more difficult to estimate using model systems. The authors should at least state these points and, in general, discuss the limitations of their study.

This point is well taken. We added to the end of the results section the following statement:

"Nevertheless, additional experiments are necessary to further optimize the molecules with respect to brain permeability and toxicity and in vivo experiments are the next crucial step to further evaluate these findings."

Reviewer #2:

Excellent revision. The authors have addressed all our concerns.

We appreciate the positive feedback on our revisions.

Reviewer #3:

The reviewers have addressed my concerns from the previous round of review and the manuscript has substantially improved.

We appreciate the positive feedback on our revisions.

1. Roskoski R, Jr. Properties of FDA-approved small molecule protein kinase inhibitors: A 2021 update. *Pharmacol Res* **165**, 105463 (2021).